Virioplankton and virus-induced mortality of prokaryotes in the Kara Sea (Arctic) in summer

Kopylov Alexander Ivanovich 1 kopylov@ibiw.ru
Zabotkina Elena Anatoliyevna 1
Sazhin Andrey Fiodorovich 2 andreysazhin@yandex.ru
http://orcid.org/0000-0002-1268-0897 Romanova Nadezda 2
Belyaev Nikolay 2
Drozdova Anastasia 2
1 Papanin Institute for Biology of Inland Waters Russian Academy of Sciences , Borok, Yaroslavl Region , Russia
2 Shirshov Institute of Oceanology of Russian Academy of Sciences , Moscow , Russia
Breitbart Mya
Electronic publication date: 2023 May 23
Publication date: 2023
Volume: 11
Electronic Location ID: e15457
Received 2022 Nov 4; Accepted 2023 May 3
Copyright: © 2023 Kopylov et al.
Copyright year: 2023
Copyright holder: Kopylov et al.
License: This is an open access article distributed under the terms of the Creative Commons Attribution License, which permits unrestricted use, distribution, reproduction and adaptation in any medium and for any purpose provided that it is properly attributed. For attribution, the original author(s), title, publication source (PeerJ) and either DOI or URL of the article must be cited.
License URL: https://creativecommons.org/licenses/by/4.0/

Keywords: Viruses, Prokaryotes, Viral infection, Kara Sea, Arctic

Funding: Russian Science Foundation 22-17-00011 The research was supported by the Russian Science Foundation project no. 22-17-00011. The funders had no role in study design, data collection and analysis, decision to publish, or preparation of the manuscript.

==============================
Among the Arctic seas, the largest volume of river runoff (~45% of the total river-water inflow into the Arctic Ocean) enters the Siberian Kara Sea. The viral communities of the Kara Sea are important for the functioning of the marine ecosystem. Studies of virus–prokaryote interactions on the Kara Sea shelf have been conducted only in spring and autumn. Here, we investigated the abundance of free viruses, viruses attached to prokaryotes, and pico-sized detrital particles; the morphology (shape and size) of the viruses, viral infection and virus-mediated mortality of prokaryotes in early summer, i.e., during a seasonal ice melting period and maximum inflow of river-water volumes with high concentrations of dissolved and suspended organic carbon. Seawater samples for microbial analyses were collected across the Kara Sea shelf zone on board the Norilskiy Nickel as a research platform from June 29 to July 15, 2018. Abundances of prokaryotes (range (0.6–25.3) × 105 cells mL−1) and free viruses (range (10–117) × 105 viruses mL−1) were correlated (r = 0.63, p = 0.005) with an average virus: prokaryote ratio of 23.9 ± 5.3. The abundance of free viruses and viral-mediated mortality of prokaryotes were significantly higher in early summer than in early spring and autumn. Free viruses with a capsid diameter of 16–304 nm were recorded in the examined water samples. Waters in the Kara Sea shelf contained high concentrations of suspended organic particles 0.25–4.0 µm in size (range (0.6–25.3) × 105 particles mL−1). The proportions of free viruses, viruses attached to prokaryotes, and viruses attached to pico-sized detrital particles were 89.8 ± 6.0%, 2.2 ± 0.6% and 8.0 ± 1.3%, respectively, of the total virioplankton abundance (on average (61.5 ± 6.2) × 105 viruses mL−1). Viruses smaller than 60 nm clearly dominated at all studied sites. The majority of free viruses were not tailed. We estimated that an average of 1.4% (range 0.4–3.5%) of the prokaryote community was visibly infected by viruses, suggesting that a significant proportion of prokaryotic secondary production, 11.4% on average (range 4.0–34.0%), was lost due to viral lysis. There was a negative correlation between the abundance of pico-sized detrital particles and the frequency of visibly infected prokaryotic cells: r = −0.67, p = 0.0008.

Introduction

Studies conducted in different Arctic regions have demonstrated that viruses constitute the most abundant component of the plankton community and play a significant role in the functioning of cold-water microbial communities, as well as in marine communities in temperate and tropical climates (Middelboe, Nielsen & Bjørnsen, 2002; Hodges et al., 2005; Wells & Deming, 2006a; Suttle, 2007; Maranger et al., 2015; Sandaa et al., 2018). In polar regions viruses maintain their infectivity at low temperatures (Middelboe, Nielsen & Bjørnsen, 2002; Weinbauer, Brettar & Höfle, 2003), and viral lysis can be important in controlling prokaryotic abundance (Guixa-Boixereu et al., 2002; Wells & Deming, 2006b). Viral lysis of prokaryotes may also influence the composition of the prokaryotic community (Weinbauer & Rassoulzadegan, 2004) and trigger the release of intracellular material upon lyses, which in turn stimulates the cycling of dissolved organic carbon (DOC) by heterotrophic prokaryotes (Bratbak, Thingstad & Heldal, 1994; Wilhelm & Suttle, 1999; Suttle, 2007). Coastal marine systems in the Arctic typically contain high concentrations of inorganic and organic particles, which enter the water column via melting of land and sea ice and runoff from large rivers (Lasareva et al., 2019; Maat, Prins & Brussaard, 2019). The high suspended particle load may substantially reduce the ability of viruses to infect prokaryotes as viruses are efficiently adsorbed by silt, clay, and organic particles (Murray & Jackson, 1992; Simon et al., 2002).

The Kara Sea is a heterogeneous and productive marine ecosystem within the Arctic Ocean, which plays a key role in the carbon cycle. The Kara Sea is mostly a shallow Arctic shelf basin influenced by river runoff: it receives 1,300–1,400 km3 of fresh water annually, accounting for 41% of total freshwater runoff to the Arctic Ocean (Makkaveev et al., 2015).

Knowledge of the patterns of the transformation of organic matter and remineralization of nutrients in the low water-temperature conditions on the Kara Sea shelf in the modern period demand detailed information on the structural and functional organization of microbial communities. These data are also necessary to assess possible future changes in the structure and functioning of prokaryotic and viral communities in the process of global warming, increased runoff from Siberian rivers flowing into the Arctic regions, and increasing anthropogenic load on the Kara Sea shelf.

In recent years, data have been obtained on the abundance of planktonic prokaryotes and viruses and virus-mediated mortality of prokaryotes in different areas of the Kara Sea shelf in early spring and autumn (Kopylov et al., 2015, 2019). However, virioplankton and viral infections of heterotrophic prokaryotes in the Kara Sea in the phenological period of early summer remain unstudied.

A specific feature of this phenological period between spring and summer is that planktonic microbial communities in the western part of the sea function under melting land and sea ice conditions, whereas in the eastern part, under conditions of maximum inflow of water masses of the Ob and Yenisei rivers, which alter the salinity regime of this water area and transport a large amount of mineral and organic matter. The river runoff of the Siberian rivers has a pronounced seasonality. In May–June, the Ob River brings 136 km3 of fresh water or 32% of the total annual river flow to the Kara Sea, and the Yenisei River—284 km3 or 45% of its annual river flow. During these two months, 1.338 × 109 g of dissolved organic carbon (DOC) or 32% of the annual DOC input enters the Kara Sea from the Ob River, 2.942 × 109 g DOC or 63% of the annual DOC input from the Yenisei River (Holmes et al., 2012). Accordingly, during this period, a much larger amount of dissolved DOC and biogenic elements enters the sea compared to other seasons (Holmes et al., 2012). Higher DOC concentrations and a relatively high water temperature, apparently, should contribute to a more intensive reproduction of heterotrophic bacteria and, accordingly, a greater activity of their obligate parasites—bacteriophage viruses.

The allochthonous dissolved and suspended organic matter entering the Kara Sea has a significant impact both on the functioning of prokaryotes and the structure and activity of viruses (Kopylov et al., 2022). The results obtained in this research can be used to fill the прокариотinformation gap on the state of planktonic prokaryotes and viruses during early summer and thereby reconstruct the seasonal cycle of the planktonic microbial community in this section of the Arctic shelf. Emerging data suggest that the planktonic microbial assemblages in the Kara Sea are active and diverse and can respond rapidly to changes in environmental conditions (Kopylov et al., 2017; Romanova & Boltenkova, 2020; Romanova et al., 2022).

Based on this research, we test the hypothesis that the abundance of free viruses is maximal in early summer, and not in other seasons: in spring; in midsummer, when the Ob and Yenisei floods end; or later, during the autumn flood of the rivers. We also assume that viral infection and virus-mediated mortality of prokaryotes is higher than in other seasons.

This determined the aim of our investigation, to assess for the first time the abundance and activity of planktonic viruses in the Kara Sea in early summer. The specific objectives were to determine (1) the abundance of free viruses and viruses attached to prokaryotic cells and pico-sized detrital particles, (2) morphological characteristics and capsid sizes of viral particles, (3) frequency of visibly infected cells and virus-mediated mortality of prokaryotes, and (4) significance of the relationship between viruses and abiotic and biotic variables.

Materials and Methods

Study sites and sampling

Water samples were collected from June 29 to July 15, 2018, on board the Norilskiy Nickel at 21 stations along the vessel’s course, from the station in the Barents Sea near the Kara Strait to the one near the Taimyr Peninsula in the Yenisei estuary. Stations were located the Marine Area (MA), a part of the shelf that receives no river runoff, and the Coastal Area (CA), adjacent to the Ob and Yenisei estuaries (Fig. 1). Samples at stations 3 and 4 were taken in ship-made channels in an ice-covered water area; stations 5, 24, and 25 were located in open water among ice fields. Other stations were ice-free.

Figure 1 The scheme of the locations of the sampling stations.

•, Circles denote stations samples were taken on June 29–July 1, 2018; ∆, triangles denote stations samples were taken on July 12–15, 2018.

Temperature was measured with an SBE-39 probe and LCD thermometer (HANNA Checktemp-1). Salinity (in practical salinity units) was measured with a Kelilong PHT-028 salinity meter (China). Surface water samples (depth 0.5 m) for biological variables and dissolved organic carbon (DOC) were collected by hand with a sterile 10-L bucket from the side of the ship. The 10-L buckets were rinsed with 0.1 M HCL prior to each DOC sampling. The DOC concentrations were measured with a Shimadzu TOC-Vcph carbon analyzer coupled with an SSM-5000A solid sample module (Belyaev, Peresypkin & Ponyaev, 2010).

Immediately after collection, water samples for microscopic studies were fixed with 25% glutaraldehyde (final concentration 1%). Samples for determining the abundance and biomass of prokaryotes were stored in sterile vials in the dark at 2–4 °C until completion of work at the station, no more than 1 h. After that, slides for epifluorescent microscopy were prepared and stored at −24 °С for 1 month before analysis.

Samples for studying viruses were subsequently stored at −80 °C until processing at an office laboratory.

Enumeration of prokaryotes and smallest organic particles

The abundances of prokaryotes (cocci and ellipsoids, rods and vibrios, and filaments were estimated separately) were determined by standard techniques using fluorochrome 4′, 6′-diamidino-2-phenylindole (DAPI) and epifluorescence microscopy (Porter & Feig, 1980). From each station, a 7-mL sample was stained with DAPI at a final concentration of 1 μg mL−1 and filtered onto a black nuclepore filter (pore size 0.2 μm). Filters were placed on glass slides and covered with Leica Type N immersion liquid and a cover glass.

Observation and counts were done under an epifluorescence microscope (Leica DM 5000 B) at a magnification of ×1,000 in two replicates. On each filter, at least 400 prokaryotes were counted and the dimensions of at least 100 cells were measured. The wet biomass was estimated based on the individual cell volume using Image Scope Color, version Image Scope M 2009, FEI Electron Optics B.V. The carbon content in prokaryotic cells (С, fg С cells−1) was calculated by the following allometric equation: С = 120 × V0.72, where V is the mean volume of prokaryotic cells, µm3 (Norland, 1993).

Yellow pico-sized organic particles (0.25–4.0 µm in size), clearly distinguished from prokaryotic cells, were also counted on DAPI-stained filters by epifluorescence microscopy (Porter & Feig, 1980; Mostajir, Dolan & Rassoulzadegan, 1995; Wells & Deming, 2003). On each filter, at least 400 pico-sized detrital particles were counted. The specific gravity of detritus particles was assumed to equal to 1.

Enumeration of viruses

The viral particles were counted under an epifluorescence microscope using SYBR Green I fluorochrome and Whatman Anodisc aluminum oxide membrane filters (pore size 0.02 μm) (Noble & Fuhrman, 1998). Depending on the viral abundance, between 0.2 and 1.0 mL of water was filtered onto the Anodisc filters. Counts were done under an Olympus BX51 epifluorescence microscope (Olympus, Tokyo, Japan) using imaging Software cellF (Olympus cell* Family; Olympus Soft Imaging Solutions GMBH, Münster, Germany) at ×1,000 magnification. For each water sample, two filters were analyzed; counts yielded a minimum of 800 viruses. The carbon content in viral particles was taken as 0.055 fg С virus−1 (Steward et al., 2007).

Viruses were also enumerated using transmission electron microscopy as described earlier (Suttle, 1993; Brum, Schenck & Sullivan, 2013). In glutaraldehyde-fixed samples, viruses, prokaryotes and smallest detrital particles contained in 50-mL samples were harvested by centrifugation onto Pioloform (SPI, Charlotte, NC, USA) and carbon-coated 400-mesh nickel grids, using an OPTIMA L-90k ultracentrifuge (Beckman Coulter, Brea, CA, USA) at 100,000 × g for 2 h. Two grids were thus prepared for each water sample.

The grids were then positively stained for 30 s at room temperature with 1% (wt/vol) solutions of uranyl acetate and lead citrate, and rinsed three times with deionized distilled water. The grids were further analyzed under a JEM 1011 electron microscope (Jeol, Tokyo, Japan) at ×50,000–150,000 magnification.

Viruses were identified on the basis of morphology (round or hexagonal capsid structures, tailed and nontailed), size and staining characteristics. Viruses were classified as myoviruses, podoviruses, siphoviruses or icosahedral nontailed viruses (referred to as nontailed viruses hereafter) based on their morphology as defined by the International Committee on Taxonomy of Viruses (Carstens, 2012). The following six classes of virus capsid size were examined to characterize viral populations: <40, 40–60, >60–100, >100–150, >150–200, and >200 nm. No less than 300 viral particles were analyzed per sample.

Transmission electron microscopy was used to measure the proportion of prokaryotic cells with attached viruses, the proportion of smallest detrital particles (0.25–4.0 µm) with attached viruses, and the abundance of viruses attached to a single prokaryotic cell and to a single detrital particle. No less than 800 detrital particles were analyzed per sample.

Virally infected prokaryotes and subsequent mortality

The method of transmission electron microscopy was used to estimate the frequency of visibly infected cells (FVIC, estimated as the share (%) of total prokaryotic abundance) and the mean number of fully matured phages in prokaryotes (i.e., burst size (BS), viruses cell−1) (Weinbauer, 2004). At least 1,200 prokaryotic cells per sample were examined to determine FVIC.

Because viruses inside prokaryotic cells become visible during the last ~10% of the lytic cycle (Proctor, Okubo & Fuhrman, 1993), FVIC were converted to the frequency of infected prokaryotes (FIC) using the equation: FIC = 7.1 × FVIC − 22.5 × FVIC2 (with data given as percentages) (Binder, 1999). Virus-induced mortality of prokaryotes (VMPR, expressed as a percentage of the production of prokaryotes,) was estimated from FIC following the model of (Binder, 1999) with VMPR = (FIC + 0.6 × FIC2)/(1 − 1.2 × FIC) and is given as a percentage. In this model, it is assumed that in a steady system, infected and uninfected prokaryotes are grazed at the same rate and that the latent period is equal to the prokaryotic generation time (Proctor, Okubo & Fuhrman, 1993; Middelboe, 2000).

Statistical analyses

Correlations between the parameters were analyzed using Spearman’s correlation coefficient calculated by Past 4.03 software (Hammer, Harper & Ryan, 2001) with regard for the prerequisites for the analyzed data.

Results

Environmental parameters

Table 1 shows values of the physical and chemical variables of the different surface seawater samples collected at stations in the Barents Sea, Kara Strait, and the Marine and Coastal areas of the Kara Sea. The water temperature and DOC values, on average for the area, were 2.7 and 3.7 times, respectively, higher in the CA than in the MA. At the same time, the average salinity and alkalinity values were 2.8 and 1.7 times lower in the CA than in the MA, respectively (Fig. 2). CA affected by the Ob and Yenisei runoff was also characterized by higher nutrient concentrations and abundance of detrital particles with a size of 0.25–4.0 µm (Fig. 2).The minimum and maximum chl a concentrations differed 150-fold in the Kara Sea (Table 1). The chl a concentration was twice as high in the CA than in the MA (Fig. 2). The chl a and nutrient concentrations were used to estimate the trophic conditions (Primpas & Karydis, 2011). Oligotrophic conditions were found most studied stations in the MA. The chl a and nutrient concentrations in the CA corresponded to oligo-mesotrophic waters.

Table 1 The physical and chemical properties of the sampling locations in the Barents sea and the Kara sea from June 29 to July 15, 2018.

Stn. name	Longitude E°	Latitude N°	Date	SIC, %	T, °C	S, psu	Alk, mg-eq L−1	NO2+NO3, μM	PO4, μM	Si, μM	NPD**, 105 particles mL−1	BPD**, mg L−1	DOC, mg L−1	
Barents sea	
1	56°29	70°18	29.06.2018	0	6.4	28.98	2.312	0.21	0.09	10.51	3.1	0.5	2.62	
25	56°68	70°22	15.07.2018	0	8.9	29.70	ND*	1.59	0.06	14.16	3.9	0.7	2.92	
Kara strait	
2	58°15	70°50	29.06.2018	0	1.2	33.42	ND	0.95	0.07	0.06	6.1	1.1	1.42	
Kara sea, Marine area	
3	60°08	70°83	29.06.2018	100	−0.7	27.12	1.789	1.56	0.10	1.76	5.9	0.9	1.39	
4	62°50	70°87	29.06.2018	100	−1.0	26.93	1.781	3.03	0.14	2.89	10.2	1.0	2.34	
5	64°30	70°92	29.06.2018	25	1.6	30.41	2.274	0.44	0.19	2.83	15.6	1.6	1.66	
6	67°68	71°79	29.06.2018	0	2.1	31.92	2.228	0.07	0.20	1.64	17.3	1.7	2.11	
7	68°46	73°02	30.06.2018	0	1.9	32.23	2.266	0.11	0.24	5.29	2.5	0.4	2.38	
8	70°82	73°74	30.06.2018	0	1.4	30.22	2.001	4.35	0.52	35.99	8.3	1.3	3.94	
21	67°85	72°14	14.07.2018	0	4.8	30.67	1.933	0.11	0.07	2.08	16.1	1.3	3.30	
22	66°92	71°55	14.07.2018	0	2.5	32.00	1.758	0.06	0.10	2.33	7.7	0.7	2.13	
23	64°08	71°01	14.07.2018	25	2.4	32.56	2.031	0.06	0.07	1.38	1.8	0.3	1.98	
24	60°78	70°80	15.07.2018	25	1.5	32.80	ND	0.09	0.13	1.01	2.2	0.2	1.62	
Kara sea, Coastal area	
9	73°57	73°89	30.06.2018	0	3.9	15.48	1.47	5.91	1.01	62.48	10.8	2.5	7.93	
10	76°06	73°88	30.06.2018	0	3.2	14.00	1.637	2.09	0.21	69.59	1.2	2.7	5.87	
11	78°06	73°54	30.06.2018	0	5.1	4.90	1.342	3.87	0.46	90.23	8.0	2.5	11.46	
12	80°13	73°00	01.07.2018	0	7.9	0.25	0.462	7.24	0.91	78.46	6.3	2.3	9.64	
17	79°52	73°27	13.07.2018	0	7.1	2.17	0.644	0.35	0.20	73.49	20.5	2.5	9.48	
18	78°14	73°68	13.07.2018	0	6.9	13.48	1.385	0.17	0.33	64.68	15.0	2.2	7.85	
19	76°00	73°80	13.07.2018	0	4.0	11.06	1.289	7.95	0.37	92.62	17.5	2.5	9.36	
20	73°11	73°79	14.07.2018	0	3.4	25.13	1.531	0.68	0.26	3.66	20.4	2.0	6.96	
Notes:

* ND, no data.

** Detrital particles 0.25 to 4.0 µm in size.

Alk, Alkalinity; BPD, mass of detrital particles; DOC, concentration of dissolved organic carbon; NPD, abundance of detrital particles; S, salinity; SIC, sea ice cover; T, temperature.

Figure 2 The average of primary physical and chemical parameters in different sections of the study area.

Average (±standard error) of temperature, T (A); salinity, S (B); alkalinity, Alk (C); NO2+NO3 (D); PO4 (E); silicate, Si (F); dissolved organic carbon, DOC (g); abundance of detrital particles 0.25–0.40 in size, NPD (H); mass of of detrital particles 0.25–0.40 in size BPD (I); in the surface water layer on the Barents and Kara seas.

Abundance of prokaryotes and pico-sized organic particles

The abundance (NPR), average cell volume of prokaryotes, and biomass (BPR) of prokaryotes varied widely in the surface water layer (Table 2). The minimum and maximum NPR and BPR values differed by 39 and 34 times across the transect, respectively. The highest values were recorded in the eastern part of the CA (stations 12, 18). The average cell volume of prokaryotes was 0.034 µm3 in the Barents Sea, 0.042 µm−3 in the Kara Strait, 0.049 ± 0.004 µm3 in MA, and 0.035 ± 0.002 µm3 in CA.

Table 2 Chlorophyll a concentration, abundance, mean volume cell and biomass of prokaryotes in the surface water layer on the Barents sea and Kara sea.

Stations	Chl a,
μg L−1	NPR,
105 cells mL−1	VPR, μm3	B PR	
mg m−3	mg C m−3	
Barents sea	
1	0.570	5.3	0.032	16.72	5.26	
25	0.621	10.0	0.035	34.53	10.59	
Kara strait	
2	0.282	0.7	0.042	2.99	0.87	
Kara sea, Marine area	
3	0.441	0.9	0.077	7.09	1.74	
4	0.883	0.8	0.057	4.65	1.24	
5	2.453	0.6	0.043	2.82	0.82	
6	3.433	1.3	0.041	5.15	1.51	
7	10.365	2.2	0.036	8.0	2.44	
8	3.823	4.6	0.035	16.13	4.95	
21	0.272	3.0	0.051	15.31	4.23	
22	0.065	1.4	0.059	8.06	2.14	
23	0.180	1.9	0.055	10.36	2.80	
24	0.318	1.7	0.033	5.56	1.73	
Kara sea, Coastal area	
9	8.742	7.4	0.021	15.79	5.59	
10	9.688	8.3	0.038	31.32	9.39	
11	2.617	8.4	0.033	27.66	8.63	
12	2.961	25.3	0.036	90.95	27.68	
17	4.418	11.8	0.037	43.26	13.07	
18	3.756	18.7	0.036	66.21	20.34	
19	6.722	7.7	0.034	26.08	8.07	
20	1.534	3.5	0.045	15.57	4.45	
Note:

Chl a, Chlorophyll a concentration; NPR, abundance of prokaryotes; VPR, mean volume cell of prokaryotes; BPR, biomass of prokaryotes.

As a result, over the entire period (June 29–July 15), NPR and BPR were on average 7.6 × 105 cells mL−1 and 7.82 mg C m−3 in the Barents Sea, 1.8 ± 0.4 × 105 cells mL−1 and 2.36 ± 0.38 mg C m−3 in MA, and 11.4 ± 2.4 × 103 cells mL−1 and 12.15 ± 2.60 mg C m3 in CA.

For the overall data set, NPR was positively correlated with T, Si, DOC, and chl a, but negatively with S and Alk. There were no significant correlations between NPR and (NO2+NO3), PO4, NPD.

The amount of pico-sized organic particles was high in the studied waters. These yellow particles are pico-sized detritus (Mostajir, Dolan & Rassoulzadegan, 1995; Wells & Deming, 2003). The abundance of detrital particles 0.25–4.00 µm (NPD) in size varied between 1.75 and 20.59 × 105 particles mL−1, and the wet weight, from 0.2 to 2.7 mg L−1 (Table 1). The average NPD value in the MA was lower than in the CA by 1.4 times, and the wet weight was 3.2 times lower (Fig. 2).

Abundance of virioplankton and composition

The epifluorescence microscopy estimates of free viral concentrations (NVF) ranged from 10 × 105 viruses mL−1 to 11.7 × 106 viruses mL−1 (Fig. 3 and Table 3). The NVF values in the Barents Sea and the Kara Strait were lower than in the MA and CA (Fig. 4). The abundance NVF in the Kara sea was on average 58.6 ± 5.7 × 105 viruses mL−1. The virus prokaryote ratio (VPR) in MA, on average 37.0 ± 7.2, was significantly higher than in the CA, on average 7.6 ± 1.7 (Fig. 4). As a result, the average VPR value on the Kara Sea shelf was 23.9 ± 5.3.

Figure 3 Electron micrographs of viruses in shelf waters of the Kara Sea.

Examples of the four viral morphotypes: (A and B) myovirus; (C–E) podovirus; (F and G) siphovirus; (H and I), non-tailed virus; (J) prokaryote with viruses on surface; (K and l) virus-infected prokaryote with viruses on surface; (M and N) viruses attached to detrital particles; (K and L) virus-infected prokaryotes—viruses inside cell observed in this study.

Table 3 Abundance of free viruses (NVF), ratio of abundance of free viruses to abundance of prokaryotes (VPR), capsid diameter of free viruses (DVF).

Station	NVF,
105 viruses mL−1	VPR	DVF, nm	
Mean ± SE	Min–max	
Barents sea	
1	37	7.0	38 ± 1	18–76	
25	36	3.7	53 ± 2	16–155	
Kara strait	
2	21	29.4	48 ± 3	17–196	
Kara sea, Marine area	
3	53	57.2	52 ± 2	23–106	
4	63	77.0	64 ± 3	26–177	
5	47	73.0	61 ± 3	16–155	
6	37	29.4	54 ± 2	21–129	
7	68	30.7	49 ± 3	26–304	
8	66	14.6	47 ± 1	16–80	
21	74	24.7	50 ± 2	16–205	
22	40	28.9	42 ± 2	16–133	
23	10	5.2	57 ± 2	21–155	
24	50	29.8	50 ± 2	16–155	
Kara sea, Coastal area	
9	38	5.2	48 ± 2	17–115	
10	51	6.2	54 ± 2	20–123	
11	103	12.3	62 ± 3	17–184	
12	117	4.6	53 ± 2	16–194	
17	57	4.8	46 ± 2	16–133	
18	82	4.4	54 ± 3	16–202	
19	36	4.7	47 ± 2	19–150	
20	63	18.2	45 ± 2	16–124	

Figure 4 The average of primary biological parameters in different sections of the study area.

Chl a, average (±standard error) of chlorophyll a (A); NPR, abundance of prokaryotes (B); NVF, abundance of free viruses (C); VPR, ratio of abundance of free viruses to abundance of prokaryotes (D); NVPR, abundance of viruses attached to prokaryotes (E); NVPD, abundance of viruses attached to detrital particles (F); FVIC, frequency of visibly infected prokaryotic cells (G), in the surface water layer on the Barents and Kara seas.

For the overall data set, NVF was positively correlated only with NPR and DOC and negatively with S (Table 4). Negative correlations were found between VPR, NPR, T, and DOC (Table 4).

Table 4 Simple (rS, Spearman rank) correlation coefficients between the abundances prokaryotes and viruses, the virus to prokaryotic ratio, frequency of visibly infected prokaryotic cells (FVIC) and environmental parameters for the whole data set.

Parameter	N PR	N VF	VPR	FVIC	
N PR		0.63**	−0.88***		
T	0.86***		−0.84***		
S	−0.73***	−0.45*			
Alk	−0.72***				
Si	0.83***				
N PD				−0.74***	
Chl a	0.48*				
DOC	0.87***	0.48*	−0.695**		
Note:

Only significant correlations are presented. Levels of significance *<0.05, **<0.01, ***<0.001. NPR, abundance of prokaryotes; NVF, viral abundance; VPR, the virus to prokaryotic ratio; FVIC, frequency of visibly infected prokaryotic cells; T, temperature; S, salinity; Alk, alkalinity; Si, silicate concentration; NPD, the abundance of detrital particles 0.25 to 4.0 µm in size; Chl a, chlorophyll a concentration; DOC, concentration of dissolved organic carbon; n = 21.

For the 6,300 viruses and 21 samples examined, only four viral morphotypes were observed: myoviruses (morphotypes with contractile tails of various shape), siphoviruses (morphotypes with long noncontractile, often flexible tails), podoviruses (morphotypes with short tails), and nontailed viruses (Fig. 3). Nontailed viruses dominated at all stations (57.0–82.0%), while podoviruses, myoviruses, and siphoviruses were the next most abundant morphotypes, in that order. The proportion of podoviruses was similar for all sites (17.3–17.9%).

The proportion of nontailed viruses out of the total viruses observed in the СА (65.9 ± 1.9%) was lower than in the МА (72.6 ± 2.0%); conversely, the proportions of myoviruses and siphoviruses out of the total viruses observed in the eastern Kara Sea were higher than those in the western, 1.6 and 2.6 times, respectively (Fig. 5).

Figure 5 Percentage (mean ± standard error) of viral morphotypes in viral assemblages in the surface water layer of the Barents and Kara seas.

The capsid diameter (DVF) of free viral particles varied from 16 to 304 nm (Table 3). The average capsid diameter varied between 37 and 64 nm per water sample, averaging 50 ± 7 nm for the 21 samples. The average DVF values in the MA and CA were close, 53 ± 2 and 51 ± 2 nm, respectively. On the Kara Sea shelf, the proportion of viruses with sizes of <40, 40–60, >60–100, >100–150, >150–200, and >200 nm out of the total virioplankton abundance was, on average, 40.54 ± 14.85%, 36.89 ± 6.21%, 18.67 ± 10.26%, 3.11 ± 2.71%, 0.64 ± 1.09% and 0.15 ± 0.37%, respectively, for all water samples. Thus, from June 29 to July 15, 2018, viruses with a capsid diameter of ≤60 nm amounted to 77.43% of the total abundance of free viruses.

The abundance of prokaryotes with viruses attached to their cells (NPRV) varied from 0.2 × 105 cells mL−1 to 11.2 × 105 cells mL−1, on average 1.6 ± 0.5 × 105 cells mL−1, which was 10.9–40.7%, on average 24.0 ± 1.4%, of the total abundance of prokaryotes (Fig. 3 and Table 5). There were from one to 12 viral particles attached to the surface of a cell of prokaryotes. From 1.2 ± 0.1 to 1.9 ± 0.3 viruses cell−1 were on the surface of a bacterial cell on average per water sample. The abundance of viruses attached to prokaryotes (NVPR) varied between 0.1 × 105 and 10.3 × 105 viruses mL−1, averaging (1.6 ± 0.1) × 105 viruses mL−1. The average NVPR value in the CA was seven times higher than in the MA (Fig. 4). The capsid diameter of viruses attached to prokaryotes varied from 16 to 167 nm. The average capsid diameters of viruses attached to prokaryotes per water sample varied between 45 and 88 nm, averaging 61 ± 0.4 nm for all samples (Table 5).

Table 5 Characteristics of prokaryotes with attached viruses and viruses attached to prokaryotes.

Station	NPRV,
105 cells mL−1	NPRV/NPR, %	NVPR/NPRV, viruses cell−1	NVPR, 105 viruses mL−1	DVB, nm	
Mean ± SE	Min-max	
Barents sea	
1	1.0	18.20	1.5 ± 0,9	1.5	50 ± 1	29–74	
25	1.2	12.50	1.2 ± 0.5	1.5	62 ± 1	46–93	
Kara strait	
2	1.4	19.38	1.6 ± 1.3	2.2	50 ± 1	24–68	
Kara sea, Marine area	
3	0.2	23.5	1.5 ± 0.8	0.3	60 ± 3	22–90	
4	0.2	26.9	1.4 ± 1.2	0.3	78 ± 1	65–101	
5	0.2	30.5	1.5 ± 0.8	0.3	62 ± 1	47–79	
6	0.3	22.9	1.5 ± 0.7	0.4	63 ± 1	36–89	
7	0.7	32.6	1.9 ± 1.4	1.4	70 ± 2	41–139	
8	1.0	20.9	1.5 ± 1.0	1.4	56 ± 2	22–77	
21	0.7	22.3	1.7 ± 1.2	1.1	66 ± 2	43–97	
22	0.3	24.7	1.6 ± 1.1	0.5	45 ± 2	27–97	
23	0.2	10.9	1.7 ± 1.3	0.4	47 ± 1	31–63	
24	0.4	26.5	1.5 ± 0.8	0.7	65 ± 1	42–92	
Kara sea, Coastal area	
9	2.1	28.1	1.6 ± 1.1	3.4	88 ± 2	55–120	
10	1.6	19.0	1.4 ± 0.8	2.2	64 ± 2	39–113	
11	2.2	26.3	1.3 ± 0.6	2.9	71 ± 3	25–147	
12	10.3	40.7	1.5 ± 0.7	15.4	69 ± 4	27–169	
17	3.3	27.9	1.4 ± 0.6	4.6	50 ± 1	36–66	
18	5.2	27.9	1.6 ± 1.0	8.3	58 ± 2	37–96	
19	1.6	21.2	1.3 ± 0.5	2.1	61 ± 2	32–110	
20	0.7	20.8	1.5 ± 0.7	1.1	49 ± 1	32–67	
Note:

NPRV, the abundance of prokaryotes with attached viruses; NPRV/NPR, proportion of prokaryotes with attached viruses of the total abundance of prokaryotes; NVPR/NPRV, abundance of viruses on the surface of a single prokaryotic cell; NVPR, abundance of viruses attached to prokaryotes; DVPR, average capsid diameter of viruses attached to prokaryotes.

The abundance of pico-sized detrital particles with attached viruses (NPDV) ranged between 0.7 × 105 to 4.2 × 105 particles mL−1 (on average 2.0 ± 0.2 × 105 particles mL−1) and was from 7.1% to 75.0% (on average 28.6 ± 3.5%) of NPD (Fig. 3 and Table 6). The average amount of NPDV in the Barents Sea and in the MA was two- and 1.4-fold lower than in CA, respectively. From one to 17 viruses were attached to the surface of a single detrital particle. As a result, the abundance of viruses attached to detrital particles (NVPD) in the Kara Sea was, on average, 5.4 ± 0.8 × 105 viruses mL−1. The average abundances NVPD in the Barents Sea and in the MA were 3.4 and 2.0 times lower than in the CA, respectively. The capsid diameter of viruses attached to detrital particles ranged from 21 to137 nm. The average capsid diameters of viruses attached to PD (pico-sized detrital particles) per water sample were 25–72 nm, averaging 56 ± 3 nm for all samples (Table 6).

Table 6 Characteristics of detrital particles with attached viruses and viruses attached to detrital particles.

Station	NPDV,
105 particles mL−1	DPD, µm	NVPD/NPDV
viruses mL−1	NVPD,
105 viruses mL−1	DVPD, nm	
min	max	Mean ± SE	Min–max	
Barents sea	
1	1.5	0.25	3.0	2.1 ± 1.5	3.2	48 ± 1	30–67	
25	1.0	0.45	3.5	1.1 ± 0.5	1.1	25 ± 1	25–45	
Kara strait	
2	2.0	0.25	4.0	2.6 ± 1.3	5.2	55 ± 2	39–84	
Kara sea, Marine area	
3	1.9	0.5	4.0	1.9 ± 1.4	3.6	72 ± 2	35–87	
4	0.7	0.3	3.0	1.4 ± 0.6	1.0	55 ± 2	41–81	
5	2.9	0.3	2.5	1.8 ± 1.2	5.2	45 ± 1	26–71	
6	4.0	0.3	2.5	2.0 ± 1.4	8.0	56 ± 2	33–79	
7	1.0	0.3	4.0	2.4 ± 1.7	2.4	69 ± 2	36–88	
8	0.9	0.5	4.0	3.2 ± 1.7	2.9	69 ± 4	40–137	
21	2.8	0.5	1.5	2.0 ± 1.5	5.6	47 ± 1	25–59	
22	2.0	0.25	2.0	2.4 ± 1.7	4.8	67 ± 2	44–111	
23	0.8	0.25	4.0	2.2 ± 1.9	1.8	51 ± 2	21–85	
24	0.9	0.3	2.5	2.4 ± 1.7	2.2	55 ± 1	38–76	
Kara sea, Coastal area	
9	2.2	0.5	4.0	2.3 ± 1.7	5.1	49 ± 1	26–63	
10	0.9	1.0	3.0	4.2 ± 2.0	3.7	70 ± 1	56–92	
11	3.4	0.3	4.0	4.3 ± 2.0	14.6	66 ± 4	25–119	
12	4.2	0.25	4.0	2.9 ± 1.4	12.2	56 ± 2	30–93	
17	2.8	0.3	4.0	3.2 ± 1.3	9.0	47 ± 1	26–75	
18	2.4	0.3	2.5	3.0 ± 1.4	7.2	47 ± 1	36–56	
19	1.9	1.0	4.0	2.0 ± 1.4	3.8	68 ± 2	34–108	
20	2.0	0.25	2.5	2.0 ± 1.2	4.0	50 ± 1	36–69	
Note:

NPDV, the abundance of detrital particles with attached viruses; DPD, the diameter of detrital particles; NVPD/NPDV, the average number of viruses on a single particle; NVPD, the abundance of viruses attached to detrital particles; DVPD, the capsid diameter of a virus attached to particles.

As a result, the total abundance of virioplankton (NVT) was (14–140) × 105 viruses mL−1, averaging 62 ± 6 × 105 viruses mL−1. Thus, the proportion of free viruses in NVT was 72.0–98.1 (on average 89.8 ± 6.0)% and was significantly higher than the proportion of viruses attached to prokaryotes, 0.3–7.6 (on average 2.2 ± 0.6)% and viruses attached to detrital particles 1.6–26.5 (on average 8.0 ± 1.3)%. The largest contribution of free viruses to the formation NVT was found at station 3 in the MA; viruses attached to prokaryotic cells, at station 12 in the CA; and viruses attached to detrital particles, at station 23 in the CA.

The total biomass of virioplankton (BVT) varied between 0.08 and 0.77 mg C m−3, averaging 0.33 ± 0.03 mg C m−3, and the proportion of virioplankton biomass of the prokaryotic biomass (BVT/BPR) varied between 2.1% and 35.0% (on average 10.1 ± 1.9%). The BVT and BVT/BPR values were, on average, 0.22 mg C m−3 and 3.2% in the Barents Sea; 0.11 mg C m−3 and 12.6% in the Kara Strait; 0.30 ± 0.03 mg C m−3 and 15.4 ± 0.9% in the MA; 0.43 ± 0.06 mg C m−3 and 4.2 ± 0.3% in the CA.

Viral infection and virus-mediated mortality of prokaryotes

The frequency of visibly infected prokaryotic cells (FVIC) in the Kara Sea ranged from 0.4% to 3.5% NPR, averaging 1.3 ± 0.2% NPR (Fig. 3 and Table 7). The average FVIC values in the МА and СА were close or slightly less than those in the Kara Strait and Barents Sea (Fig. 4). There was no significant positive correlation between NPR and FVIC. A negative correlation was found between the abundance of NPD and FVIC (Table 4).

Table 7 Frequency of visibly infected prokaryotes and infected prokaryotes, virus-mediated prokaryotic mortality, number of phages inside cells and ratio of the number of infected cells to the number of cells with attached viruses.

Station	FVIC,
% оf NPR	FIC, % of NPR	VMPR, % of PPR	BS, viruses cell−1	NPRVIC/NPRV, %	
Mean ± SE	Max	
Barents sea	
1	2.2	14.5	19.2	5.7 ± 0.1	8	11	
25	1.4	9.0	10.6	10.0 ± 0.4	15	18	
Kara strait	
2	1.8	12.0	15.1	6.0 ± 0.2	8	20	
Kara sea, Marine area	
3	1.9	12.7	16.1	7.0 ± 0.2	11	17	
4	1.3	8.8	10.4	6.0 ± 0.2	9	28	
5	1.4	9.0	10.6	6.0 ± 0.3	11	28	
6	0.4	2.8	2.9	4.0	4	147	
7	3.5	22.1	34.0	5.0 ± 0.2	9	14	
8	1.2	8.2	9.5	5.8 ± 0.3	10	26	
21	0.8	5.5	6.1	6.0 ± 0.2	8	56	
22	0.8	5.5	6.1	6.3 ± 0.1	8	40	
23	1.2	8.2	9.5	17.0 ± 0.7	24	30	
24	1.2	8.2	9.5	5.5 ± 0.1	7	42	
Kara sea, Coastal area	
9	1.0	6.9	7.8	6.3 ± 0.2	9	40	
10	2.5	16.3	22.3	9.4 ± 0.8	32	10	
11	1.4	10.1	12.2	15.2 ± 1.1	35	26	
12	1.4	10.1	12.2	7.0 ± 0.3	11	37	
17	0.7	4.9	5.4	5.0 ± 0.1	6	28	
18	1.0	6.9	7.8	5.3 ± 0.1	6	31	
19	1.0	6.9	7.8	5.3 ± 0.1	7	11	
20	0.7	4.9	5.4	5.4	6	26	
Note:

FVIC, the frequency of visibly infected prokaryotic cells; FIC, the frequency of infected prokaryotic cells; VMB, the virus-mediated prokaryotic mortality; BS, the number of mature phages inside prokaryotic cells; NPRVIC/NPRV, the ratio of the abundance of infected cells to the abundance of cells with attached viruses, %.

Calculations based on the FVIC estimates showed that the proportion of virus-infected cells of NPR (FIC) varied from 2.9% to 22.1% of NPR (on average 9.2 ± 0.9% of NPR), and the viral-mediated mortality of prokaryotes (VMPR) was 4.0–34.0% (on average 11.4 ± 1.5%) of the prokaryotic production. The abundance of visibly infected prokaryotic cells (NPRVIC) was (1–36) × 103 cells mL−1, on average, 8 ± 2 × 103 cells mL−1. The minimum and maximum NPRVIC/NPRV ratios differed by seven times (Table 7) and the NPRV/NPRVIC ratio varied from 8 to 58. A strong positive correlation was found between NPRV and NPRVIC, r = 0.89, p < 0.005, n = 21. Whereas the average abundance of viral infected prokaryotic cells in the MA was six times lower than in the CA, the NPRVIC/NPRV values in these areas did not differ significantly, 5.8 ± 1.0% and 4.9 ± 1.1%, respectively.

The number of phages in viral-infected prokaryotic cells (BS) fluctuated from 4 to 35 phages cell−1, averaging 7.1 ± 0.7 phages cell−1 (Table 7).

Cocci+ellipsoid cells (58–71%) made the main contribution to the total abundance of prokaryotes; rods and vibrios (28–41%) and filaments (0–3.0%) were less abundant (Fig. 6). Rods and vibrios accounted for the largest fraction of virus-infected prokaryotes (69–94% of the total abundance of infected prokaryotes), with lower numbers observed for cocci, ellipsoids, and filaments (6–28% and 0–3%) (Fig. 6). That is, the phages infected prokaryotic cells of different morphology at a different rate. The proportion of virus-infected rods and vibrios in the total abundance of prokaryotes with the corresponding morphology was highest in the Barents Sea. The proportion of virus-infected cocci and ellipsoids in the total abundance of prokaryotes with the corresponding morphology was lowest in the CA.

Figure 6 Proportions of prokaryotic cells of different morphology in the total number of prokaryotes and in the total number of infected cells.

Percentage (%) of prokaryote cells of various morphology to the total prokaryote abundance (A); percentage (%) of infected prokaryote cells of various morphology to the total number of infected prokaryote cells (B), percentage (%) of infected prokaryote cells of various morphology to the number of prokaryote cells of a given type (C).

Percentage of infected cells to the number of cocci and ellipsoids in the MA was higher than in the CA, but the percentage of infected cells to the number of cocci and ellipsoids in the MA was lower than in the CA (Fig. 6).

Discussion

Abundance and biomass of viruses

In the current study, a high abundance of free viruses (NVF) was detected in the Kara Sea in early summer. NVF consistently correlated with the abundance of prokaryotes (NPR), whereas other physical, chemical, and biological parameters correlated only weakly or not at all with NVF. Other researchers have also observed that virus-like particles were significantly correlated with bacterial abundance, but correlations with other physicochemical or biological parameters were insignificant (Stopar et al., 2003). The typical viral life cycles (lytic and lysogenic) and replication rates are closely linked with host metabolism. Consequently, the factors regulating the physiology of the host, as well as its production and removal are also important in governing virus dynamics (Mojica & Brussaard, 2014; Zhang, Weinbauer & Peduzzi, 2021). A strong positive correlations between NVF and NPR suggest that a significant amount of bacteriophages are present in the virioplankton (Steward, Smith & Azam, 1996; Wommack & Colvell, 2000; Auguet et al., 2005).

Comparison of the results of studies of planktonic free viruses on the Kara Sea shelf conducted at different times of the year showed that in in early summer, the average NVF 58.6 ± 5.7 × 105 viruses mL−1 and VPR 23.9 ± 4.9 values (present study) were higher than those obtained in early spring, 10.8 ± 0.2 × 105 viruses mL−1 and 6.9 ± 0.8 (Kopylov et al., 2019) and in autumn, 17.3 ± 0.8 × 105 viruses mL−1 and 5.0 ± 0.5 (Kopylov et al., 2015; Kopylov et al., 2017). The NVF in the Kara Sea are within the range of NVF (0.1–64.1) × 106 viruses mL−1 and VPR 0.8–70.0% values recorded in the central Arctic Ocean and other Arctic seas (Steward, Smith & Azam, 1996; Hodges et al., 2005; Steward et al., 2007; Clasen et al., 2008; Venger et al., 2016). In early summer, in Kara Strait and Kara Sea of viral biomass (BV) was 10.7 ± 4.7% of prokaryotic biomass (BPR) (present study). In early spring, the BV: BPR ratio was significantly lower, 2.2 ± 1.3% (Kopylov et al., 2019). For comparison, the viral biomass in the central Arctic Ocean was about 6% of the prokaryotic biomass (Steward et al., 2007).

Morphological characteristics

Only four morphotypes (myoviruses, siphoviruses, podoviruses, and nontailed viruses) were observed in this study, indicating that other morphotypes (for example, lemon-shaped or filamentous) made up <1% of these marine viral assemblages. Nontailed viruses appear to dominate numerically in the Kara Sea, since they constituted, on average, 65.9–72.6% of the viral particles observed. Similar to the results obtained in other marine habitats (Stopar et al., 2003; Auguet, Montanie & Lebaron, 2006), Brum, Schenck & Sullivan (2013) found that nontailed icosahedral viruses dominate in the upper water column of the global oceans, making up 51–92% of viral assemblages.

The majority of pelagic viruses in this study were less than 60 nm in diameter. Similar results were reported for estuarine and marine waters at different latitudes (Bergh et al., 1989; Cochlan et al., 1993; Weinbauer & Peduzzi, 1994; Alonso et al., 2001; Auguet, Montanie & Lebaron, 2006). In Kara Sea waters, the capsid size of viruses attached to prokaryotic cells rarely exceeded 100 nm, 61 ± 2 nm on average. Viruses of eukaryotic algae typically have a larger capsid size (100–180 nm, on average 152 nm) (Van Etten, Lane & Meints, 1991), whereas the majority of icosahedral nontailed phages observed had head sizes <70 nm, making it likely that these could be bacteriophages (Hanson, Berges & Young, 2017).

In summer 2018, the proportion of viral particles with a capsid diameter <100 nm (mainly bacteriophages) of NVF (96%) is higher than in spring 2016 (80%); conversely, the proportion of viruses with a capsid diameter >100 nm (hosts of which are mainly algae and other eukaryotic organisms) is significantly lower in summer (4%) than in spring (20%). As a result, the average capsid diameter of viruses in summer (51.7 ± 1.3 nm, present study) was lower than in spring (79.8 ± 1.3 nm (Kopylov et al., 2019)). The presence of a large number of phycoviruses in Kara Sea shelf waters in early spring 2016 was probably due to the end of the diatom bloom on the lower surface of ice. This was evidenced by both the appearance of the lower edge of the ice, colored brown, and the remains of characteristic colonies of ice algae (mainly Nitzshia frigida Grunow, 1880) in surface water samples. In addition, during this period, the bloom of Phaeocystis pouchetii (Hariot) Lagerheim, 1896 had begun, already forming numerous colonies (Sazhin et al., 2017).

Viral infection and virus-mediated mortality of prokaryotes

In Arctic waters, the FVIC and VMPR values most often vary from 0.5% of NPR and 3.7% of PPR (central Arctic region (Steward et al., 2007)) to 2.1% NPR and 20.2% of PPR (coastal waters of the Novaya Zemlya archipelago (Venger et al., 2016)). Thus, in summer, the average FVIC and VMPR values in surface waters on the Kara Sea shelf are in the middle of the range of values determined in different regions of the Arctic (Steward, Smith & Azam, 1996; Middelboe, Nielsen & Bjørnsen, 2002; Boras et al., 2010).

As is known, the frequency of contacts between viral particles and prokaryotic cells depends on their respective abundance, the physical and chemical parameters of the water, and the size of a given prokaryotic cell and given viral capsid (Murray & Jackson, 1992). It is possible that the relatively larger size of rods and vibrios (from 0.6 × 0.2 to 1.7 × 0.6 μm) and filamentous prokaryotes (from 1.1 × 0.3 to 2.7 × 0.8 μm) compared to the cocci and ellipsoids cell (from 0.3 × 0.2 to 1.0 × 0.6 μm) contributes to a higher frequency of contacts between these morphological types of prokaryotes and viruses and, as a consequence, to the higher probability of viral infection of these prokaryotes.

A high concentration of PD with less than 4 µm in size was found in surface waters of the Kara Sea shelf; their abundance exceeded that of prokaryotes. A large amount of PD with attached viruses was detected. The average abundance of viruses attached to pico-sized PD exceeded fivefold the average abundance of viruses attached to prokaryotes, suggesting that adsorption of viral particles to pico-sized PD reduced both the abundance of free viruses and the level of viral infection of prokaryotes. A negative correlation was found between the abundance of PD and FVIC r = −0.67, p = 0.0008.

By adsorption to nonliving organic particles, the viruses are thus, at least temporarily, unavailable for infection of new host cells, and adsorption of viruses to organic particles is expected to have a profound inhibitory effect on virally mediated mortality of microorganisms (Brussaard, Kuipers & Veldhuis, 2005; Mojica & Brussaard, 2014).

The low abundance of viruses during the transitional period from spring to summer is explained by adsorption of viruses to suspended inorganic particles entering Arctic coastal waters with runoff from adjacent land and to detrital particles formed in large quantities after the phytoplankton bloom (Schoemann et al., 2005).

Recently, it was experimentally demonstrated that different virus populations strongly adsorb to fine-grained glacial sediments. Moreover, production of progeny was strongly delayed in the presence of glacial sediments (Maat, Prins & Brussaard, 2019; Maat, Visser & Brussaard, 2019).

In the CA, the abundance of prokaryotes was 6.3 times higher than in the MA. At the same time, the abundance of free viruses in the CA was 1.4 times higher than in МА. A higher concentration of suspended particles in waters of the eastern Kara Sea may be one of the possible reasons for the relatively low concentration of free viruses in the CA, as well as in some other marine habitats (Hewson & Fuhrman, 2003; Maat, Visser & Brussaard, 2019). According to our data, the weight of detrital particles with a size of 0.254 µm on average for the area is three times higher in the CA than in the MA. The lowest (0.2–0.5 mg L−1) suspended particulate matter (SPM, particle size from 0.47 to 1 mm) were detected in the western Kara Sea, whereas the SPM concentration reaches 10 mg L−1 in the eastern part (Burenkov, Goldin & Kravchishina, 2010). Based on our data, the average abundance of viruses attached to detrital particles less than 4 µm in size were twice as high in the CA as in the MA. It was shown earlier that the low viral abundance generally observed in the presence of high suspended particle load might be caused by adsorption to suspended particles (Simon et al., 2002). We hypothesize that, higher concentrations of suspended particles entering the CA in large amounts with Ob and Yenisei runoff and, correspondingly, higher adsorption of viruses to these particles resulted in the reduced abundance of free-living viruses and hence a lower specific contact rate between prokaryotes and viruses, decreasing the frequency of visibly infected prokaryotic cells.

Conclusions

In late June–early July (i.e., at the end of the phenological spring–early summer), viruses are an essential component of the planktonic microbial community on the Kara Sea shelf, averaging about 10% of the prokaryotic biomass. The abundance of free viruses in very early summer was higher than those found on the Kara Sea shelf in early spring and autumn. A strong positive correlation was found between the abundance of prokaryotes and free viruses. Free viruses significantly prevailed in the total abundance of virioplankton. Nontailed viruses quantitatively constituted the majority of free viruses in Kara Sea water samples. The predominant viral particles had a capsid diameter of 16–60 nm.

A large number of viruses were attached to detrital particles with a size of 0.25–4.0 µm. The negative correlation between the frequency of virus-infected prokaryotic cells and the abundance of pico-sized detrital particles suggests that the latter are an important factor reducing the level of viral infection of prokaryotes. According to the obtained values of viral-mediated mortality of prokaryotes, in early summer, viruses in general play an appreciable role in controlling the abundance of prokaryotes on the Kara Sea shelf.

Supplemental Information

Supplemental Information 1 Physical and chemical and biological parameters at sampling stations.

The average values and standard errors.

Click here for additional data file.

The authors would like to thank S. I. Metelev, G. Bykov, and Z. Bykova of the Centre for Electron Microscopy, Papanin Institute for Biology of Inland Waters, Russian Academy of Sciences, for their help in preparing material for electron microscopy. The authors are deeply grateful to Dr. Aaron Carpenter for correcting the English translation. The authors are also grateful to three anonymous reviewers for their thorough work with the manuscript.

List of Abbreviations

PD Pico-sized detrital particles

NPD Abundance of pico-sized detrital particles

NPDV Abundance of pico-sized detrital particles with attached viruses

NPR Abundance of prokaryotes

NPRV Abundance of prokaryotes with attached viruses

VPR Volume of prokaryotic cell

BPR Biomass of prokaryotes

NVF Abundance of free viruses

VPR Virus: prokaryote ratio

NVPR Abundance of viruses attached to prokaryotic cells

NVPD Abundance of viruses attached to pico-sized detrital particles

DVF Capsid diameter of free viruses

FVIC Frequency of visibly infected prokaryotic cells

NPRVIC Abundance of visibly infected prokaryotic cells

FIC Frequency of infected prokaryotic cells

VMPR Viral-mediated mortality of prokaryotes

BS Burst size

Additional Information and Declarations

Competing Interests

Author Contributions

Data Availability

The authors declare there are no competing interests.

Alexander Ivanovich Kopylov conceived and designed the experiments, performed the experiments, analyzed the data, prepared figures and/or tables, authored or reviewed drafts of the article, and approved the final draft.

Elena Anatoliyevna Zabotkina conceived and designed the experiments, performed the experiments, analyzed the data, prepared figures and/or tables, authored or reviewed drafts of the article, and approved the final draft.

Andrey Fiodorovich Sazhin conceived and designed the experiments, performed the experiments, analyzed the data, authored or reviewed drafts of the article, and approved the final draft.

Nadezda Romanova performed the experiments, analyzed the data, authored or reviewed drafts of the article, and approved the final draft.

Nikolay Belyaev performed the experiments, authored or reviewed drafts of the article, and approved the final draft.

Anastasia Drozdova performed the experiments, authored or reviewed drafts of the article, and approved the final draft.

The following information was supplied regarding data availability:

The raw data is available in the Supplemental File.

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
