# Peer review of "Virioplankton and virus-induced mortality of prokaryotes in the Kara Sea (Arctic) in summer"

_PeerJ, doi:10.7717/peerj.15457_

## Round 0.1 · original submission · Major Revisions

Thank you for your submission. Three reviewers have provided feedback on the manuscript and made suggestions on how to improve the manuscript. In particular, please improve the clarity of the introduction to state clear goals and/or hypothesis of the study and put the work into context with prior literature. This will make the manuscript less descriptive in nature and can help you highlight interesting and novel findings. All three reviewers felt there was a lot of data but the implications of the findings were not well described (why does it matter, what have you learned through this study). The reviewers also noted that including tables/figures on all of the measured parameters, incorporating more metadata if available (or can be derived from satellite information), and performing statistical analyses comparing the study locations would enhance the manuscript. Finally, there are some methodological issues that could use further clarification in order to allow full evaluation of the results.

Reviewer 1 ·

Basic reporting

The manuscript is reasonably well written. However, there are several areas where the writing and accuracy of the text can be improved. For example. There are a lot of typos
the manuscript would benefit greatly from an introduction that is more focused on building an argument that clearly identifies a current gap in our understanding about virus mediated mortality of prokaryotes in the Arctic region, which is then followed by a clear set of hypotheses that will be evaluated by the manuscript to help fill in this gap in understanding.
For example.
The central aim of the study identified in lines 98-100 is to characterize the virioplankton community during the early summer. But why is this important? Why summer? What factors do you believe will be important in influencing the abundance of free and attached viruses?
Regarding the secondary aim of the study specified in lines 101-104 to evaluate differences in the western and eastern part of the sea? I assume from the limited text of the introduction that the central belief is that differences will arise due to suspended particles associated with river runoff? However, salinity can also impact the abundance, absorption, and infectivity of marine viruses, how will the impact of these factors be disentangled? If this is indeed particle load is believed to be the driving factor, the study would have benefited from including a measure of suspended particles, or turbidity, as well some indication of particle composition (as particle chemistry can have a large influence on their ability to bind to and impact viral dynamics).
In addition, there too many abbreviations that deviate from the typical abbreviations found in literature. This makes it very hard to follow along in the text without deviating to the list of abbreviations.

The manuscript would also benefit from improvements in consistency and accuracy of presented results. For example, in the text and table abundances are presented as per mL but Figure 4a lists per L. In addition, there are several typographical or spelling errors in the axis labels of Figure 4.

Lines 219. Inconsistent units of cell volume.

The manuscript would also benefit from presenting the important trends in their data in a more succinct and effective manner, as in its current state it is very hard to follow.

Experimental design

The research question and underlying hypothesis of the study are not well defined.

As mentioned previously, the manuscript would benefit from a well-defined question that is able to be addressed using the data presented/collected.

In addition, the manuscript may benefit from additional statistical analysis. For example, cluster analysis or multivariate analysis such as PCA could be used to support differences in the physicochemical environment between the different locations within the Kara Sea.

According to lines 160-161. Viruses were identified based on morphology. But no trends in morphology were presented. Morphology characteristics might provide more useful information than size classes. Viral morphology is central to viral taxonomy, aspects of their biology (and may be particularly relevant to their sensitivity to salinity) and have been documented to vary across environmental gradients. It seems like a huge, missed opportunity to not include this data in this study.

There are several places where the detail and clarity of the methods can be improved. For example.

Lines 126-127. I am unfamiliar with the term ‘luminescent microscopy’. Is this fluorescence microscopy?

Lines 127. I am also unfamiliar with storage at -24°C

Lines 128. Details on the fixation method for viruses needs to be added.

How was the association or attachment of a virus to a particle characterized or distinguished?

Lines 194-196. There were results from statistics presented that are not described here. For example. The t-test stated in Line 231. In addition, I am unfamiliar with t-test results stating a ‘t’ range.

Validity of the findings

The manuscript seems to lack the sufficient data to support the conclusions presented. This can be improved by (1) increasing the clarity and focus of the introduction and stating clear hypotheses to be testing, (2) using the available data more effectively (i.e., viral morphology data, and statistical analysis demonstrating difference between study locations, (3) including additional data (i.e., suspended solids, turbidity –if not information not collected can obtain for the time of the study from satellite data) and (4) more effectively presenting trends in the data in a manner that makes it more palatable to the reader.

Reviewer 2 ·

Basic reporting

The authors describe for the Kara Sea the abundanes of viruses and prokaryotes, the % visibily infected cells, the size of the virus particles, the number and size of Dapi-yellow stained particles and estimates of prokaryotic biomass and the mortality of the prokaryotes by viruses. Despite that the combination of abundances with viral size and organic particle load is an interesting one, the results are presented in a descriptive manner without a clear hypothesis or focus which make it rather difficult to follow.
Some of the interesting results such as the organic particles (number and size) are not listed in a table or presented as images. Perhaps authors can compare / focus on the organic particle load results with the prokaryotic host and virus abundances for the different stations to improve on the scientific novelty.
Several of the stations are located very close together (e.g., station 10&19, 11&18, 9&20, 5&23, 1&25) but the authors failed to address this aspect and compare those stations in relation to the dates they were sampled and potential changes in physicochemical characteristics. It seems large variation in prokaryotic abundances and biomass can be found between these closely situated stations. Without addressing this aspect, a comparison of the current summer cruise findings with the literature spring findings becomes difficult.

Experimental design

Methods are suited but authors should make sure that all necessary details are provided. For example, electron microscopy counting of viruses is mentioned (line 159) but no details how the samples were fixed and processed are provided, company information is typically missing, references are missing occassionally (e.g., FVIC electron microscopy method), depth of sampling is not provided (surface, meaning 0m? or 0.5m? or 3m?), nor are the dates or the coordinates of the different stations sampled.
Can the authors provide more information about the stations, e.g., Chlorophyll concentration as source of DOM and inorganic nutrient concentrations?
Were the 10L bucket containers for sampling DOC acid rinsed prior to use?
Samples for prokaryote and virus enumeration were fixed and left at 2-4C for at least several hours before further processing. That is a risk as fixed samples are 4C have been shown to show rapid decline in virus abundances. This is not a constant value (differs for different viruses and different sample sites), which may have affected the results.
Authors used electron microscopy to determine the % viruses attached to cells and organic particles but there are no details on how the samples were processed. If centrifugation was involved, how can the authors be sure that the attached viruses were not passively adsorbed to particles as a result of sample handling?

Validity of the findings

It is recommended to clarify some of the methodological issues (see comments above) as well as present some of the currently missing data (volume of prokaryotes, organic particle abundance and size) in tables and images to improve the robustness of the conclusions. The variation between closely positioned stations should be taken into account.
Authors assume that virus mediated mortality of prokaryotes equals the losses of prokaryotic production, but why is grazing not taken into account? (the authors did consider grazing on infected and uninfected cells).
Suggest to place Nvf in Table 1 and keep EM vs TEM abundances and size in separate table.
Nvf/Npr is similar to the generally used VPR (virus to prokaryotic host ratio); for clarity it is recommended refer to VPR.
Line 235: comparing 2-3 days sampling period without listing which stations that involves (this information is also not provided earlier in manuscript) is scientifically not helpful. And why compare only those 2 three-day periods while the cruise was from 29 June to 15 July?
The discussion starts with comparison summer and spring as if both seasons were sampled during the present study. Same again later on when discussing virus sizes. Make sure the reader understands that you compare to published data and provide reference. Btw, is the difference in virus capsid size presented for summer and spring significant? Especially considering the variation between closely situated stations sampled twice within one month (current study). Please also acknowledge there might have been overall variation between years (summer was 2018 and spring 2016).
Line 361-7: how do the findings relate to the trophic state of the different geographical locations?

Additional comments

Authors use many abbreviations but MA and CA can easily be spelled out each time used to enhance readability. Please check if this holds for some of the other abbreviations.
Might be helpful to present figures of the data that correlate significantly (currently only mentioned in the text).

Reviewer 3 ·

Basic reporting

The goals of the study were clear: more characterization of the study sites have occurred in early spring and autumn, less is known regarding the early summer melt. Expect interactions between virioplankton and suspended particles (inorganic & organic) in meltwater; expect differences in bacterioplankton and virioplankton communities in terms of abundance, morphological composition, in east-west transect (riverine influence to marine influence).

The manuscript reads well and sufficient background is provided to gauge context. Sound data reporting in tabular form.

Figure 1 is of very low quality and is not acceptable as-is. Please reconstruct with higher resolution.

Fig 2. All examples of prokaryotes with viruses on the surface of the cell also were visibly infected. I am not sure that this is a meaningful category if the virus attached to the surface of the cell mediated the infection. I think it may be more meaningful to report the proportion of prokaryotes with attached virus particles, but no evidence of intracellular viral particles.

Figure 3 – should the caption specify that this is free viruses? (I assume it is free viruses) . l.396 states that viral abundance did not differ significantly across these zones, whereas Fig. 3a suggests that viral abundances were quite different based on sampling location. Please change the language in l. 396 to reflect this.

Experimental design

l.101 - I am not sure what is meant by "differences in the structure of virioplankton" - do you mean *community* structure? And if so, how will such differences be determined?

I have several questions that should be answered or clarified in order to meet standards:

L120: “sterile 10-L bucket container” – how was this sterilized? Was this re-used across stations or multiple separate, sterile containers, one for each sample/station?

l. 125 – 127: please clarify how samples were handled start to finish. Approx. how many hours of storage between sample collection and (assumed) slide preparation? I was initially confused about the phrase “preparations to be examined to luminescent microscopy were prepared” – I was wondering what specific steps were taken, and in what form the samples were stored. This was later specified on l. 135, but to me, it would make sense to include these details on preparation here, when they are first mentioned. Also, why not store these samples at -80 C along with viruses?

l. 141: please provide a manufacturer/publisher (and version number, if applicable) for the Image Scope Color software that is mentioned here.

l. 148: “smallest” is subjective – what was the size (diameter or volume) range of the discernible organic aggregates that were included in this portion of the analysis?

l. 155: Cell F Image Analysis software – please specify the manufacturer/publisher (and version number, if applicable) for this software.

l. 161: was any attempt made to catalogue filamentous morphologies? It seems like you are potentially missing a large group of viruses here. Why not include these observations?

l. 163: could you provide more detail on how TEM grids were prepared? Generally, must either concentrate virus particles prior to drop-casting on grids, or samples are spun down onto grids using an ultracentrifuge. Further, different approaches are used to visualizing VLPs vs. cells, and so some details on how samples were prepared for each of these techniques would aid in evaluation and in replication of these experiments. Please include any centrifugation steps and speeds, type and concentration of stain, and contact times for samples & stains with grids.

L. 173: when were samples fixed with glutaraldehyde? Prior to storage shipboard? After samples

Approaches for FVIC determinations and associated equations are appropriate, and carried out as previously reported.

Environmental metadata (temp, salinity,DOC) are somewhat limited. This study could have included pH, chl-a, turbidity/TSS to enhance statistical analysis and identify additional relationships. What is known regarding cyanophages in these waters? While one cannot go back and perform these analyses now, I think it's worth at least acknowledging the limitations on analysis due to the omission of these measures.

Validity of the findings

Most of the results are fairly straight forward, either reporting quantitative measurements, results of calculations (e.g., FIC, viral-mediated mortality) or correlations. I had some difficulty completely following the authors' interpretations of their data & conclusions (more specifics on that below).

l. 370 – 374: I think it's worth consider alternative hypotheses, as there could be legitimate reasons other than cell dimensions that would explain this observation.

l. 380: instead of “apparently” should use “suggesting,” since the exact reasons for this are a matter of hypothesis rather than conclusion.

l. 395 – 404: it was stated the the ratio of virus to prokaryotes was lower in the CA than MA because prokaryotic abundance was higher in CA than MA, while viral abundance was about the same. This same section goes on to explore reasons why viral abundance might be lower – but hasn’t this already addressed why VBR was lower? I.e., because cell abundance was higher. Furthermore, Fig. 3a indicate that viral abundances were quite different based on sampling location. This passage needs to be reconsidered and re-written so that it is consistent with the results shown.

l. 411-413: the correlation is suggestive, not indicative, that many virus particles were bacteriophages. Morphological analysis and a preponderance of tailed particles (i.e., unequivocably phages) would be more indicative.

l. 419 - 421: I am not sure what this sentence means. Are you saying that, in spite of differences (such as abundance of organic particles) the levels of visibly infected cells and viral mediated mortality was similar between the marine and coastal areas? But if that is correct, it would negate previous statements about the presence of suspended organic particulates reducing levels of viral infection in prokaryotes. This is confusing. How would you clarify these points? Was there a statistically significant difference in particulate matter and FVIC between the coastal vs. marine areas, or not?

Reasonable comparisons with other values reported in the literature for comparable environmental samples. One of the challenges is that there are not many other comparable studies.

Additional comments

l.211-212: suggest specifying that the max/min values for prokaryotic abundance and biomass varied *across the transect*.

l. 220: Please change “High” to “Significant” positive correlations, as p-values support this designation.

l. 223: “The amount of the pico-sized organic particles was significant” – here, advise against using the term “significant” because it carries statistical connotations. Were any tests performed to determine statistical significance regarding pico organic particulate abundance? It seems like the intent is to simply say that there was high abundance. If that is the case, it can simply be stated that abundance was high.

L. 230: “EM” is a common abbreviation for electron microscopy, and TEM for, more specifically, transmission electron microscopy. What does EM mean here, since it is already specified that TEM was used in this study? If it is epifluorescence microscopy, more commonly it is abbreviated EFM.

l. 232: wouldn’t 10 x 10^5 be 1 x 10^6? I am not sure why the values are being constrained to the 10^5 range. Please consider this for all other abundance values reported.

l. 240: again, “significant” should be used rather than “high” because p-values support this description.

l. 250 and beyond: I am not sure why the numerical values are reported in parentheses. This just made it harder for me to read (more symbols to wade through and parse).

---

## Round 0.2 · Minor Revisions

Thank you for this significantly revised manuscript. One of the original reviewers has provided an additional review (which I agree with, please fix those issues) and I have provided minor suggestions below. Please address all of these comments and then the manuscript will be acceptable for publication (i.e., it will not undergo an additional round of peer review beyond my own reading).

The abstract is current composed of 3 paragraphs (one of which is only a single sentence), please combine into one concise paragraph.

Line 27: change “inflow of maximum river-water volumes” to “maximum inflow of river-water volumes”

Lines 36-37: change “viruses attached to a pico-sized detrital particle” to “viruses attached to pico-sized detrital particles”

Line 41: add % after 11.4

Line 71: In this first paragraph (in the review PDF), there are several different fonts/sizes/bold – please check for consistency.

Line 75: “Polar Regions” should not be capitalized

Line 87: Change “The Kara Sea is heterogeneous” to “The Kara Sea is a heterogeneous”

Line 114-116: Should this sentence be part of the previous or following paragraph?

Line 117: Change “we check the hypothesis” to “we test the hypothesis”

Line 162: Change “epifluorescent microscope” to “epifluorescence microscope”

Line 169: Change “Yellow pico-sized organic particles (0.25–4.0 μm in size), clearly distinguished from prokaryotic cells, were also counted, as well as prokaryotes, on DAPI-stained filters by epifluorescence microscopy.” To remove the “as well as prokaryotes” clause so it reads “Yellow pico-sized organic particles (0.25–4.0 μm in size), clearly distinguished from prokaryotic cells, were also counted on DAPI-stained filters by epifluorescence microscopy.

Line 173: “The specific gravity of detritus particles was taken equal to 1.” – is this an assumption? If so, say “The specific gravity of detritus particles was assumed to equal 1.”

Line 184 and throughout the manuscript: when saying “fgС”, there should be a space “fg C”

Line 204: Edit to simplify language – I propose “Transmission electron microscopy was used to measure the proportion of prokaryotic cells with attached viruses, the proportion of smallest detrital particles (0.25−4.0 μm) with attached viruses, and the abundance of viruses attached to a single prokaryotic cell and to a single detrital particle.”

Line 256: “0.042 μm-3 in the Kara Strait” should be μm3 (without the “-“)

Line 259: For “mgC” put a space “mg C”

Line 265: I’m not sure how you determined that these are organic particles (as opposed to inorganic particles)? I understand that they don’t stain with DAPI, which binds nucleic acid, so that distinguishes them from cells.

Line 288: “Podoviridae as a proportion of total viruses observed was similar for all sites (17.3–17.9%).” – but Podoviridae no longer exists as a viral family (see latest ICTV release), so I would suggest to just say “The proportion of podoviruses was similar for all sites (17.3-17.9%).”

Line 291: Do not capitalize “Myoviruses and Siphoviruses”

Line 321: Replace “fluctuated between” with “ranged from”

Line 322: PD is introduced here but not defined until line 450 – please define at first use on line 322.

Line 328: “The largest contribution of free viruses to the formation ща” – doesn’t make sense, may be a typo at the end of this line, please fix.

Line 352: Instead of “high positive correlation” it might be better to use “strong positive correlation” (same on line 380 and 494)

Line 358: Replace “Cocci+ellipsoid cells (58–71%) made the main contribution to the formation of the total abundance of prokaryotes” with “Cocci and ellipsoid cells (58–71%) made the main contribution to the total abundance of prokaryotes”

Line 378: should be “virus-like particles”

Line 382: remove the word “composition”

Line 388: add “105 viruses mL-1” after “6.9±0.8”

Line 394: I find the wording confusing in “In early summer, the average proportion of virioplankton biomass (BV) of the prokaryotic biomass (BPR) in Kara Strait and Kara Sea was 10.7±4.7% (present study).” – would it be accurate to say “Virioplankton biomass (BV) was 10.7±4.7% of the prokaryotic biomass (BPR) in the Kara Straight and Kara Sea (present study)”? I still find it a little confusing because virioplankton biomass wouldn’t be part of the prokaryotic biomass, would it? Would it be better to say it was ten-fold lower than prokaryotic biomass?

Line 407: Reword this to say “Similar to the results obtained in other marine habitats (Stopar et al., 2003; Auguet, Montanie & Lebaron, 2006), Brum, Schenck & Sullivan (2013) found that nontailed icosahedral viruses dominate in the upper water column of the global oceans, making up 51–92% of viral assemblages.” – I removed the first citation to Brum, Schenck & Sullivan (2013) and then tightened up the second citation.

Line 419: “Cyanophages” are mentioned here for the first time, but unless you also made measurements of the proportion of cyanobacterial to total prokaryotic populations, would suggest removing this and just saying “it likely that these could be bacteriophages”.

Line 446: Replace “cocci+ellipsoids cell” with “cocci and ellipsoid cells”

Line 451: Remove “the” in “exceeded the that of”

Line 459: Replace “have had” with “have”

Line 462: Here you introduce inorganic particles – can you expand on why you think your particles are organic and whether you expect that to make a difference in viral adsorption?

Line 482: Replace “Probably,” with “We hypothesize that”

Line 492: I’m confused by this sentence: “The abundance of free viruses was higher than of those found on the Kara Sea shelf in early spring and autumn, i.e., in very early summer.” Would it be correct to say “The abundance of free viruses in very early summer was higher than those found on the Kara Sea shelf in early spring and autumn”?

Please carefully proofread the table and figure legends. For example, Table 4 should be “Simple (rS, Spearman rank) correlation coefficients between the abundances of prokaryotes and viruses, the virus to prokaryote ratio, frequency of visibly infected prokaryotic cells (FVIC) and environmental parameters…”

Reviewer 1 ·

Basic reporting

Overall the revised manuscript has been improved both in overall flow and writing. There are still several issues that should be addressed before publication. These include an increased clarity/accuracy of reported results, and additional information about statistical methods applied to data.
The authors need to be mindful of significant figures (particularly in tables)
Table 4 - authors should include all correlations, not only significant ones (and be mindful of significant figures and typos)

Experimental design

Regarding the use of correlation analysis - did the authors apply any correction for multiple comparisons (for example Holm's or Bonferroni) to control for family wise error?

Procedures for filtration methods typically do not use the phrase 'pour onto a filter' (ex. ln 159, 179, etc.) - material is typically filtered onto a x um pore sized filter or concentrated over a x um pore sized filter

Validity of the findings

no additional comments

Additional comments

Ln 107-109; This sentence requires a reference or needs to be restated in a less definitive manner
Ln:116; Hypotheses have both a cause and effect - your hypothesis is that the abundance of free viruses is maximum in the early summer ---Why?
Ln254&258; still some inconsistency in reporting units
Ln302: Not sure why it would be necessary to report the average per water sample
Ln322: typo after the word formation
Ln 361-362: 'occurrence frequency' is a redundant statement as occurrence is the frequency
Ln: 372 virus like
Ln: 374 there are also studies that show that some physicochemical variables are significantly correlated with viral abundance and viral production
Ln 375 'are' present and remove 'composition'
Ln 387:'proportion of virus biomass of prokaryote biomass' does not make sense. please check the wording there
Ln 388: same comment here - role of viruses in the 'formation' of biomass' needs to be reworded

---

## Round 0.3 · accepted · Accept

Thank you for carefully addressing the reviewer's comments. I have assessed the revision myself and believe that the revised manuscript is ready for publication